



# One year of high frequency monitoring of groundwater physico-chemical parameters in the Weierbach Experimental Catchment, Luxembourg.

Karl Nicolaus van Zweel[1,2], Laurent Gourdol[2], Jean François Iffly[2], Loïc Léonard[2], François Barnich[2], Laurent Pfister[2], Erwin Zehe[3], and Christophe Hissler[2]

[1]VTT Technical Research Centre Ltd, P.O. Box 1000, FI-02044 VTT, Finland
[2] Luxembourg Institute of Science and Technology (LIST), Avenue des Hauts-Fourneaux 5, L-4362 Esch-sur-Alzette, Luxembourg
[3]Karlsruher Institut für Technologie (KIT), Institut für Wasser und Gewässerentwicklung, Lehrstuhl für Hydrologie, Otto-Amman-Platz 1, 76131 Karlsruhe

**Correspondence:** Christophe Hissler (christophe.hissler@list.lu)

**Abstract.** The critical zone (CZ) is the Earth's skin where rock, water, air, and life interact - playing a pivotal role in sustaining ecological processes and life-supporting resources. Understanding these interactions, especially in forested headwater catchments, is key for managing water resources, predicting environmental responses, and assessing human impacts. We present a novel dataset from the Weierbach Experimental Catchment in Luxembourg, derived from a year-long high-frequency moni-
toring campaign that focused on groundwater physico-chemical parameters. Through meticulous data collection and rigorous quality control, parameters such as electrical conductivity, dissolved oxygen, oxidation-reduction potential, and pH were measured, offering new insights into the CZ's hydrological and biogeochemical dynamics. The dataset highlights the intricate interplay between redox reactions, pH, and ion exchange processes, as well as the influence of seasonal variability and flowline interactions on solute transport. By providing a detailed view of the catchment's response to hydrological changes, this dataset
fills a significant gap in CZ research, offering a valuable resource for advancing our understanding of hydro-biogeochemical catchment processes. The datasets introduced in this contribution may be leveraged by researchers and practitioners aiming to refine models, inform land management practices, and foster a more holistic understanding of catchment biogeochemistry.

## 1   Introduction

The critical zone (CZ) stands as the interface where rock, water, air, and living organisms interact, playing a vital role in supporting ecological processes and providing essential resources (Amundson et al., 2007; Anderson et al., 2012; Brantley et al., 2007). The dynamics within this "skin of the Earth" are fundamental for managing water resources, predicting environmental responses to disturbances, and assessing the impacts of human activities, especially against the backdrop of climate change.



Within the CZ, the fields of hydrology and biogeochemistry interconnect deeply, offering insights into ecological sustainability and resource management (Wymore et al., 2022).

The subsurface properties, especially variations in permeability with depth, exert a significant control on the vertical connectivity within the CZ, affecting water flow and chemical interactions. For example, in systems where vertical permeability decreases sharply, most water travels through only the top layers of the subsurface, limiting its interaction with deeper, reactive rock formations. (Xiao et al., 2021). This results in a pronounced chemical contrast between shallow and deep waters, where shallow water, often younger and flowing rapidly through high-permeability soil layers, carries fewer geogenic solutes compared to the older, deeper water that has interacted more extensively with bedrock (Welch and Allen, 2014; Jin et al., 2010; Brantley et al., 2013). These differences are highly relevant as they shape the solute export patterns observed at catchment outlets, influencing the concentration vs. discharge (c-Q) relationships. Understanding these subsurface interactions is important, as they not only determine stream chemistry but also provide insights into hydro-biogeochemical processes, which are essential for assessing ecological sustainability and managing water resources effectively.

One of the most accessible methods to understand the interactions occurring within the CZ is by observing catchment outlets. Streams, acting as integrators of all processes within the CZ, reveal the cumulative effects of these interactions on stream behaviour and hydrochemical composition, thus serving as a window into the overall dynamics of the CZ (Brantley et al., 2017; Li et al., 2017; Riebe et al., 2017). This approach underpins the importance of analysing concentration vs. discharge (c-Q) patterns, which have been a focal point of research, providing insights into the hydrological and biogeochemical functioning of catchments (Brooks et al., 2015; Zhi et al., 2019; Stewart et al., 2022). This line of inquiry underscores the complex interplay of hydro-biogeochemical processes within the CZ, often masked by spatial heterogeneities. Despite advancements, field measurements and their interpretations remain challenging, with water physico-chemistry data reflecting convoluted signatures that obscure direct cause-and-effect relationships (Weiler and McDonnell, 2006). Recent consensus, emerging from diverse measurements and observations, points towards the determination of stream chemistry by the properties of source waters, predominantly from shallow subsurface and/or deeper regolith and fractured bedrock flow paths. The contrasting chemical properties of these source waters are pivotal in shaping solute export patterns (Stewart et al., 2022).

Given the complexity of interactions within the CZ, direct data supporting the nuanced understanding of subsurface and stream physico-chemical interactions are rare. This research gap emphasises the need for detailed datasets that can capture the intricacies of CZ dynamics. Addressing this need, our study presents a novel dataset centred on the Weierbach Experimental Catchment (WEC) in Luxembourg. Chosen for its extensive prior research and status as a CZ observatory, the WEC offers an unparalleled opportunity to delve into the hydrological and biogeochemical processes shaping the CZ (Hissler et al., 2021).

This paper introduces our methodology for data collection, focusing on the significance of subsurface architecture knowledge and the exploration of hydrological dynamics through the dataset. By presenting this dataset, we aim to contribute to the broader understanding of complex hydro-biogeochemical processes within the CZ at specific subsurface locations within a long-term experimental catchment. This effort not only enhances our comprehension of water resource dynamics but also underscores the importance of integrating hydrological and biogeochemical data for environmental monitoring and management.



## 2 The Weierbach Experimental Catchment (WEC)

The catchment is located in the northwestern region of Luxembourg (Lat: 49.8273; Long: 5.7956) and has an area of 45 ha. The WEC is a forested headwater catchment within the Alzette River basin (Pfister et al., 2017). The catchment is located in the Ardennes massif, which is distinguished by an elevated subhorizontal plateau and deep V-shaped valleys. It is subdivided in three landscape units (Figure 1). The plateau area is about 31 ha and has slopes between 0° and 5°. The hillslope area is about 14 ha and has slopes between 5° and 44° (Martínez-Carreras et al., 2016). The riparian area represents about 0.4 ha. The catchment's stream network consists of three tributaries that converge into a single stream and is bordered by a highly dynamic riparian zone composed primarily of litter degradation material and colluvium from the hillslopes (Antonelli et al., 2020a, b). The riparian area in the catchment is narrow, flat, and well-defined with some areas where the riparian zone widens.

The WEC's forest structure is shaped by past and current management practices. Oak trees are evenly distributed, while beech trees' density increases from plateau to footslope. The primary plateau has a dense mixed forest with 78% European beech trees and 22% oak hybrids (Fabiani et al., 2022), where the understory is dominated by blueberries, and the riparian zone has various short plants (Martínez-Carreras et al., 2015). The secondary plateau is primarily Norway spruce and Douglas fir.

The WEC experiences a semi-marine climate with even precipitation distribution throughout the year, averaging 953 mm annually (2006–2014; Pfister et al. (2017)). Streamflow varies with seasons, with lower base flow during July to September due to higher evapotranspiration losses (annual average PET of 593 mm for the period of 2006-2014, from Pfister et al. (2017, 2023)). In dry conditions, intermittent streamflow occurs from upstream to downstream, characterized by sharp, short-lasting discharge peaks (Wrede et al., 2015). In wet conditions, a double peak behaviour is observed, with a rapid discharge peak shortly after rainfall, followed by a broad, long-lasting second peak that outweighs the first peak in volume (Martínez-Carreras et al., 2016; Scaini et al., 2018). Figure 2 displays the daily total precipitation in the WEC and the daily mean discharge at SW1 from late 2008 to 2022.

The WEC regolith is composed of two parts called solum, where pedogenic processes are dominant and where biota play an important role and subsolum, where the original rock structure or fabric of the bedrock is preserved (Juilleret et al., 2016). The structure of these two parts evolve according to the topography. The WEC's solum between 0 and 70 cm consists of a stony silt loam soil derived from periglacial solifluction deposits that is overlain on a weathered Devonian slate substratum. From the surface, a thin, dark organic surface horizon (O to A horizon, 0-8 cm on average) is followed by a cambic horizon (B horizon, 8-40 cm on average) with a well-developed angular blocky soil structure. The texture of the silt loam is inherited from the loess deposit and was mixed with slate clasts via solifluction, with the quantity of clasts increasing with depth. The subsolum is composed by slates, phyllites and quartz of Devonian age (Moragues-Quiroga et al., 2017). During the last 10 years, extensive drilling and geophysics (Gourdol et al., 2021) campaigns has been conducted. The WEC's deep subsurface exploration yielded more detail than is currently known. The upper part of the catchment can be subdivided between a primary plateau where the solum developed over a deep yellow-white slate saprolite and a secondary plateau where the solum directly developed over the black fractured slate bedrock. These distinct landscape units are separated by the black line in Figure 1, which represents the weathering front. The spatial occurrence of these two distinct regolith structures was observed using combined core drilling

descriptions and electrical resistivity survey (Gourdol et al., 2021). In the hillslope and riparian areas, the subsolum is composed on fractured and/or fresh bedrock underneath the solum. A mineralogical and geochemical characterization of the WEC's regolith was conducted to gain a deeper understanding of the primary plateau regolith characteristics and evolution. This study

allowed observing three distinct compartments that have different geochemical – and mineralogical characteristics as a function of contrasting evolutions. These respective processes are related to different atmospheric deposition events, weathering ages and seasonal water saturation dynamics (Moragues-Quiroga et al., 2017).

## 2.1 Data Description

### 2.1.1 Fiveteen years of groundwater monitoring

The WEC has been thoroughly equipped to continuously monitor water fluxes and physico-chemical parameters within various CZ compartments. Additionally, these compartments are sampled every two weeks at different locations to analyze the $\delta18O$ and $\delta2H$ isotopic composition of water, as well as its chemical composition, including rainfall, throughfall, groundwater , and streamwater. Some of these datasets have already been published (Hissler et al., 2021). The WEC monitoring infrastructure is continuously expanded and now encompasses an extensive network of 26 monitoring boreholes with varying depths, ranging

from 1.95 meters to as deep as 40 meters below the surface. All the boreholes are spatially distributed all over the catchment (Figure 1) and have screens that cover the range from 1 meter below the surface to the bottom. For this study, 9 boreholes were selected along one hillslope catena, representing one unique vegetation type, but several hydrogeological positioning. The chosen boreholes include locations on the primary plateau (GW1 and GW5), in the middle of the hillslope (GW7 and GW9) and at low hillslope positions (GW2, GW10-12 and GW3). Groundwater levels in these boreholes are automatically

recorded every 15 minutes using OTT Orphimedes and CTD (OTT, Aix-en-Provence, France). The relevant data are shown in Table 1. The depth reference for the water sampling is the top of the pipe, whereas it is the surface of the soil for the screen of the pipe. The standpipe represents the difference between the top of the pipe and the surface of the soil.

### 2.1.2 One year of physico-chemical parameters monitoring

A weekly frequency monitoring campaign was designed with the objective of achieving higher resolution concerning key po-

110 tential processes expected to occur in the subsurface of the catchment. The duration of the campaign spanned from 02/03/2021 to 14/03/2022, during which sampling was conducted at weekly intervals. The parameters measured during each sampling event included electrical conductivity (EC), dissolved oxygen (DO), oxidation-reduction potential (ORP), corrected for probe bias to Eh, measures the water's redox potential using a platinum electrode, commonly expressed relative to the standard hydrogen electrode (SHE) for comparison, and pH, all of which were analysed using a flow-through cell system (Figure 3). The

115 four parameters were recorded using two WTW® Multi 3630 IDS data loggers connected to the specific probes: the WTW® TetraCon® 925 probe for EC, the WTW® FDO® 925 probe for DO, the WTW® SenTix® ORP-T 900 probe for ORP, and the WTW® SenTix® 940 probe for pH. The pH probe underwent calibration in the field at each sampling event using HACH Singlet™ Solution Packs for a two-point calibration process (pH1=4.01 and pH2=7.00). In addition to the four parameters, the





study also recorded pumping time, water level before and after pumping, as well as the quantity of water extracted from the well.

Prior to measure the parameters and sampling a borehole, a five-minute period was allowed for the probes to record the EC datum reading, which ranged from 0.1 to 1.7 $\mu$S/cm, after flushing the system with one litre of demineralized milliQ water. Subsequently, the borehole was pumped, and the four parameters were continuously recorded at one-minute increments. A groundwater sample was collected when the EC reading remained stable for several minutes, under the assumption that the borehole demonstrated homogeneity. After collecting the water sample, a 20-minute equilibration period was implemented, wherein the flow-through system was sealed, and the probes were allowed to equilibrate with the solution in the cells. The final readings for EC, DO, ORP, and pH were then obtained. The physico-chemical database consists of three subsets:

- The first subset includes a high frequency continuous pumping test at borehole GW5 (Figure 1 and Table 1). These tests were conducted twice: once during the winter and again during the summer, covering both wet and dry seasons. During these tests, the four physical parameters (EC, DO, ORP and pH) were measured at 10-second intervals, while water samples were taken every minute for measuring hydrochemistry.

- The second subset comprises time series data representing the four parameters obtained from weekly sampling events. In this dataset, each sample site is evaluated weekly for four physicochemical parameters: pH, EC, ORP, and DO. The alignment of the time series data for all physico-chemical parameters was based on the EC values. Consequently, the peaks in EC for each time trend, corresponding to their respective sample sites, may not always align due to variations in the time required for the probes to reach equilibrium after flushing the flow-through cell setup with the demineralized milli-Q water. To standardise the data across various sampling sites, a centring process was employed. This process entailed the identification of the highest EC value within each time series. Upon locating this maximal EC value, the time scale of the dataset was subsequently adjusted. Specifically, the time series was recalibrated such that the instance corresponding to this peak EC value now serves as the reference starting point for each series. This adjustment ensures consistency in the time frame across all measurements, thereby facilitating more precise comparisons and analyses of the data trends.

- The third subset comprises weekly physical parameters, which were measured after a 20-minute EC equilibration time. It also contains related chemical analysis data from the water samples collected during these weekly events, along with additional metadata recorded during the sampling process. This metadata includes information such as the water level of the well before pumping, the water level of the well after pumping, the drawdown, and the volume of water removed during each pumping event. Furthermore, this subset includes the daily mean water level obtained from the level logger data.

### 2.1.3 Water sample preparation and analysis

All water samples were stored at 4°C prior to be prepared for the different chemical analyses. They were filtered in the laboratory during the day following the collection. A vacuum system and 0.45 $\mu$m cellulose acetate membrane filters, manufactured



by Sartorius (type 111, 47 mm diameter), were used. During pre-processing of the samples EC was also measured at the laboratory using a TetraCon® 925 probe. On the filtered water samples, the pH and alkalinity were measured using a Mettler Toledo equipment with 0.01N HCl up to pH 4.5; the dissolved silica ($SiO_2$), the ammonium ($NH_4^+$) and the orthophosphate ($HPO_4^{3-}$)

concentrations and the absorbance at 254 nm (Abs254) were measured using spectrophotometry with a Skalar© continuous flow analyzer SAN++; the major cations ($Na^+$, $K^+$, $Ca^{2+}$, $Mg^{2+}$) and anions ($Cl^-$, $NO_3^-$, $SO_4^{2-}$) concentrations were analyzed using ionic chromatography (Dionex ICS-5000 dual channel). On the filtered and acidified (1% $HNO_3$) water samples, the total content of major metals (Al, Fe, Mn, Ba, Sr) were analysed with a Quadrupole ICP-MS (Agilent 7900) associated with an ISIS 3 (Agilent) injection system. The analyses were conducted in He mode and $^{103}Rh$ and $^{185}Re$ were used as internal standards.

The calibration standards were prepared with Multi elements ICP standard solutions (CHEMLAB Analytical) diluted in 1% $HNO_3$. The analytical blank values were less than 1% of the lowest sample concentrations for all elements. For all analyses, the quantification limits are given in Table 2.

### 2.1.4 Data Quality

This study implements a comprehensive data quality control approach throughout its year-long physico-chemical monitoring

campaign, ensuring the reliability and accuracy of the data collected. Parameters such as EC, DO, ORP and pH were precisely measured using standardized instrumentation, adhering to established protocols for environmental monitoring. Regular calibration of our equipment was conducted before each data collection to ensure measurement accuracy. This involved both factory standards and field adjustments to account for environmental variables specific to the WEC. Meticulous sampling procedures were followed, complemented by established analytical techniques, to maintain methodological integrity. The dataset's struc-

tured organization, complete with detailed metadata, enhances its transparency and usability for further research. Acknowledging potential data variability challenges, such as probe equilibration times which could be influenced by seasonal changes, we implemented procedures to ensure consistency across all measurements. We invite data users to engage with us for validation of specific data subsets and to discuss any discrepancies, a step critical for maintaining the dataset's quality and integrity. These protocols collectively underscore the dataset's robustness, advocating for its proper citation and use in line with the highest

standards of scientific inquiry.

### 2.1.5 Data Usage and Applications

When analysing c-Q patterns in forested headwater catchments, it is vital to understand the dynamics of biogeochemical processes taking place in the shallow horizons of the system such as redox reactions or cation and anion exchanges in soil. Indeed, the varying patterns of c-Q relationships observed in different catchments can be explained by the switching dominance

of different end-member source waters, which are driven by from subsurface biogeochemical heterogeneity (Zhi et al., 2019; Stewart et al., 2022). Our dataset potentially affords the capacity to quantify the relative contributions stemming from both shallow and deep flowlines during borehole pumping. This leverage is predicated upon the intrinsic course of electron acceptor succession (McMahon and Chapelle, 2008). Consequently, this phenomenon instigates the formation and stratification of redox zones. As such, when borehole pumping is conducted contingent upon the hydrological state of the system, the composition

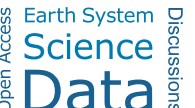

of the extracted mixture can be computed based on the relative redox potential (Eh) of said mixture. The provided database allows to address Gaps in the Current State of the Art on:

- **High-Resolution Monitoring**: While previous studies have explored c-Q relationships and the influence of various water sources on stream chemistry, the novelty of this dataset lies in its high-resolution monitoring. By collecting data at frequent intervals, including the weekly measurement of four physical parameters, it offers a level of detail that allows for a more nuanced understanding of dynamic catchment processes. This high-resolution approach addresses a gap where data may be limited to lower-frequency measurements. The temporal behaviour, at different timescales, observed in GW5 is shown to illustrate the dynamic interplay between shallow and deep flowlines and how this influences the solute behaviour observed in the groundwater (Figure 4 and Figure 5). This borehole was investigated in more detail and at higher frequency during two sequential sampling experiments conducted in high flow and low flow. It can be observed from these two graphs that the mixing ratio between the two flowlines (shallow and deep) changes as a function of the season in the catchment. The relationship between parameters that drive biogeochemical processes (EC, DO, ORPEh, pH – top of Figure 5) and the solute concentration ($Ca^{2+}$, Alkalinity, $SO_4^{2-}$ and $NO_3^-$ - bottom of Figure 5) can also be inferred and useful for new hydrological and biogeochemical modelling approaches.

- **Quantifying Redox Reactions**: The dataset significantly contributes to addressing a key gap in current knowledge related to the quantification of redox reactions within the shallow horizons of forested catchments. While previous research has acknowledged the importance of redox processes, this dataset provides the necessary empirical data to model and quantify these reactions accurately. As an example, understanding how the dissolved organic carbon (DOC), precipitation, and subsequent redox reactions affect the catchment's chemical composition represents a substantial advancement. In natural shallow systems, water can only be oxidized/reduced under extremely oxidizing/reducing conditions, and the actual lower and upper Eh boundaries are governed by $CO_2$ reduction to $CH_4$ and the presence of $O_2$, respectively. Using Eh values, one can determine the dominant redox couple in a system. Because multiple electron donors can drive multiple redox reactions at the same time, systems rarely have strict redox boundaries in practice. The redox state of the soil and the unsaturated zone can vary greatly (Mansfeldt, 2003). It can be oxidizing, weakly reducing, moderately reducing, or strongly reducing ($\geq 400$, $200 - 400$, $100 - 200$, and $-100$ mV, respectively). Figure 6 shows the in-situ pH vs. Eh measurements taken throughout the one-year weekly campaign in the different boreholes. The defined zones, based on the redox zoning in soils (Zhang and Furman, 2021), indicate how the redox state of the soil and the unsaturated zone can vary greatly. Only boreholes GW1 and GW11 are seen to exhibit weakly reducing behaviour, with GW1 exhibiting a preference for moderately reducing conditions during the summer period. This behaviour can be understood, considering that GW11 represents the deepest currently observable flowline in the system (Figure 1-b), while GW1 corresponds to shallow groundwater on the plateau. GW1 represents vertically infiltrating water, influenced by redox processes associated with shallow and biologically active zones in the system.

- **Influence on pH and Ion Exchange**: The dataset's ability to capture the interplay between redox reactions and pH changes extends our understanding of CZ processes. Current research often focuses on individual aspects of the system,





but this dataset enables a more holistic view by elucidating how redox conditions can influence pH and, consequently, the exchangeable pool of ions sorbed to the soil. This integrated approach addresses a gap in the current literature, which tends to treat these processes in isolation. Under steady-state chemical conditions, the composition of a cation exchanger will be the same as that of the groundwater around it. When the composition of the water changes due to acidification, or change in vegetation influence, the cation exchanger readjusts its composition to the new groundwater concentrations. As a result, the exchanger serves as a temporary buffer that can completely change the concentrations in water via a process known as ion chromatography (Appelo and Postma, 2004). The main sources of acidity in the most near-surface environments are $CO_2$ from the atmosphere and organic acids formed by decaying DOC, resulting in an acid pH range of 5–6. Acid sulfate soils have lower pH values, with values as low as 4 occasionally recorded (DeLaune and Reddy, 2005). The lower bound for pH in natural shallow environments can then be assumed to be this low pH value.

– **Seasonal Variability and Flowline Dynamics**: One of the dataset's unique strengths is its capacity to elucidate the seasonal variability and flowline interactions within the catchment. Previous studies may lack the temporal granularity needed to capture these nuances adequately. By demonstrating how the dominance of flowlines changes with varying hydrological conditions, the dataset offers insights into hydrological dynamics that were previously underexplored in the literature. In Figure 7, boreholes GW3, GW7, GW9, GW10, and GW12 are grouped together as representing shallow flowlines. The shaded area in the graphs reflects the median trend of field parameters (indicated by the dashed grey line), along with the 95% confidence interval derived from these five boreholes (depicted as a shaded grey area). On the other hand, the black line represents the deep flowline (GW11). The behaviour of both GW5 and GW2 is also depicted individually. It's evident that the dominance of one flowline over the other changes depending on the hydrological state of the system. Lastly, the behaviour of GW1 is also presented. It's apparent that this borehole is primarily influenced by chemical processes related to the consumption of dissolved oxygen and subsequent changes in redox conditions.

– **Practical Implications**: Our findings on the relationship between stream chemistry and subsurface water chemistry have practical implications. They suggest that stream chemistry can serve as a valuable proxy for assessing and monitoring subsurface water chemistry. This bridges a gap in the current state of the art by highlighting the potential utility of readily available stream chemistry data for inferring subsurface conditions. To illustrate the respective influences of the shallow and deep flowlines on the temporal trend in the stream water at the outlet (labelled SW1 in Figure 1), a graph has been provided in Figure 6, where these influences are depicted in blue. This visual representation helps clarify how different flowlines contribute to changes in stream water chemistry over time.

## 3 Conclusions

The one-year monitoring campaign conducted within the WEC has generated a rich dataset that extends beyond conventional hydrological and chemical monitoring efforts. By capturing the complex relationships among physical parameters, water chemistry, and redox reactions, this dataset empowers researchers to more deeply explore the intricate processes governing redox



zonation within the catchment. Moreover, its insights into the influence of redox reactions on pH and ion exchange widen our comprehension of how these interactions influence the catchment's biogeochemical landscape. As a dynamic dataset that incorporates both seasonal variability and flowline interactions, it emerges as a crucial tool for advancing our knowledge of CZ processes in forested headwater catchments. Weekly campaigns were executed to collect data on four physical parameters, including EC, DO, ORP, and pH from a comprehensive network of 10 sample points encompassing nine boreholes and one stream water. The temporal trends observed in DO and ORP through this monitoring campaign provide valuable insights into the intricate processes affecting redox conditions within the groundwater system. However, the true strength of this dataset resides in the continuous measurement of these physical parameters coupled with the concurrent collection of water samples at each monitoring point. This synergistic approach significantly enriches our understanding of the dynamic water quality characteristics within the WEC. The novelty and significance of this dataset are multifaceted:

- **Modelling Redox Reactions in Shallow Horizons**: The dataset offers essential parameters for accurately modeling redox reactions within shallow horizons, particularly highlighting the critical role of DOC in catalyzing chemical changes during precipitation events. The detailed documentation of the series of reactions initiated by the influx of DOC—its oxidation, the subsequent release of $CO_2$, formation of carbonic acid, and the dissolution of bicarbonate minerals—provides a comprehensive view of the catchment's chemical responses to rainfall, offering a nuanced understanding of the biogeochemical processes at play.

- **Influence of Redox Reactions on pH and Ion Exchange**: This dataset sheds light on the significant impact of redox reactions on pH levels and their potential cascading effects on the soil's ion exchange capacity. It allows for an in-depth examination of the complex relationship between redox changes and pH variations, which is vital for deciphering the broader implications of these shifts on the catchment's nutrient and solute dynamics, including the availability and mobility of essential nutrients and solutes.

- **Hydrological Variability and Flowline Interactions**: The dataset uniquely captures the catchment's seasonal variability, emphasizing the intricate dynamics between shallow and deep flowlines and their role in solute distribution. The detailed temporal analysis reveals how the interaction between these flowlines evolves in response to seasonal variations, providing key insights into the hydrological patterns and solute transport mechanisms that govern the catchment's ecological and chemical processes.

In conclusion, this dataset not only enriches the existing body of knowledge but also contributes to overcoming prevailing knowledge gaps. Through its high-resolution monitoring, emphasis on redox reactions, detailed exploration of pH dynamics, and comprehensive insights into seasonal and flowline dynamics, it substantially enhances our comprehension of CZ processes in forested headwater catchments. This is a substantial contribution for the scientific community, offering researchers and practitioners a robust foundation to refine predictive models, guide land management strategies, and foster a more integrated approach to catchment hydro-biogeochemistry. By leveraging this dataset, the scientific community is poised to make significant strides in ecological conservation and resource management, contributing to a more sustainable and informed stewardship of natural landscapes.

*Data availability.*  The database related to this data paper is publicly available at zenodo.org (https://zenodo.org/records/10869166) (Hissler et al., 2024). All datasets are cleaned with outliers removed. A detailed description of all monitoring and sampling sites, including physiographic characteristics, field equipment details and sampling procedures, is presented in all shared .xlsx files. The hydrological timeseries for each borehole and the discharge at the outlet are given in a separate file. Weekly timeseries of EC, DO, ORP and pH and the sample hydrochemistry are given in separate files. For more information on the available dataset, please communicate with the corresponding author
(christophe.hissler@list.lu).

*Author contributions.*  Karl Nicolaus van Zweel collected the majority of the field data as part of his PhD research and contributed to the manuscript writing. Laurent Gourdol and Christophe Hissler provided significant assistance in data collection and manuscript preparation. All three authors were involved in data processing. Jean François Iffly was responsible for field installations of equipment and provided additional data. Loïc Léonard conducted laboratory analyses and data capture, with guidance from François Barnich. Laurent Pfister and

Erwin Zehe offered technical input, reviewed the manuscript, and contributed to its writing. The collaborative efforts of all authors were invaluable in the completion of this manuscript.

*Competing interests.*  The authors declare that they have no conflict of interest.

*Acknowledgements.*  This work is part of the HYDRO-CSI project and was supported by the Luxembourg National Research Fund (FNR) in the framework of the FNR/PRIDE research programme (contract no. PRIDE15/10623093/HYDRO-CSI/Pfister). We would like to thank
Jérôme Juilleret, Viola Huck, for their contributions to the water sample preparation.





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

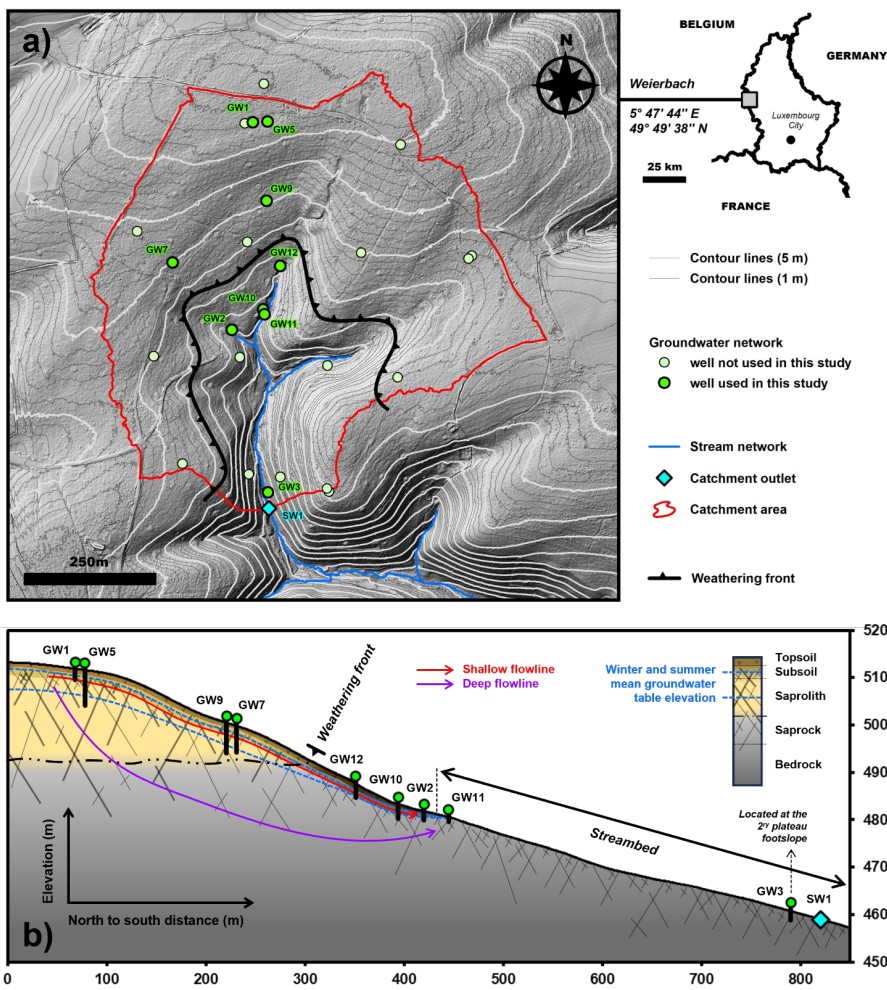

**Figure 1.** Presentation of the Weierbach Experimental Catchment (a) topography and water sampling location and (b) regolith structure. The black contour line in the map (a) marks the chemical weathering front basement derived from ERT surveys and core drillings (Gourdol et al., 2021).

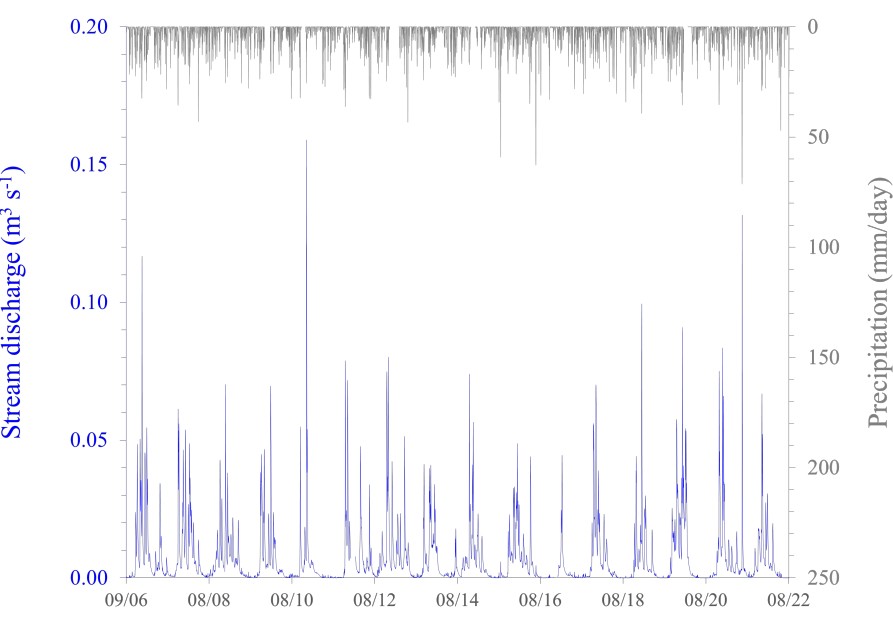

**Figure 2.** Daily total precipitation in the Weierbach Experimental Catchment and the daily mean discharge at the catchment outlet from 2006 to 2022.

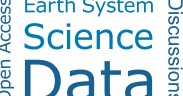

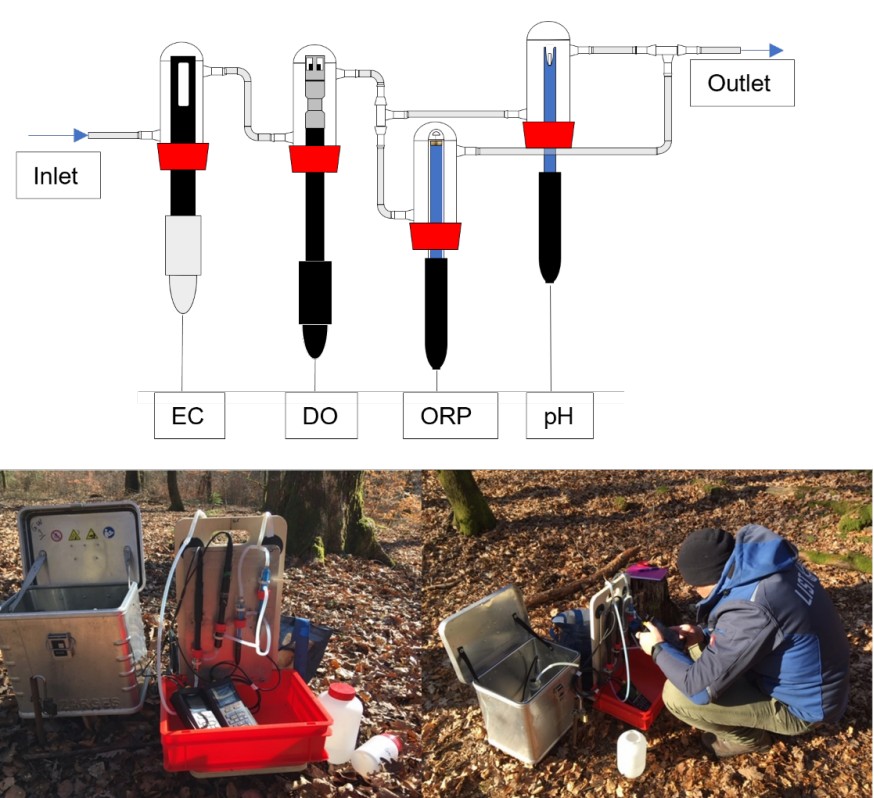

**Figure 3.** The schematic at the top illustrates the flow-through cell used to measure in-situ electrical conductivity (EC), dissolved oxygen (DO), oxidation-reduction potential (ORP), and pH. The ORP and pH probes were placed in parallel to avoid interference. Two WTW® Multi 3630 IDS data loggers were used to automatically record the probe data. The two photos at the bottom left and right show the setup used in the Weierbach Experimental Catchment and its field application.

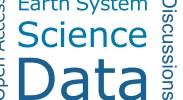

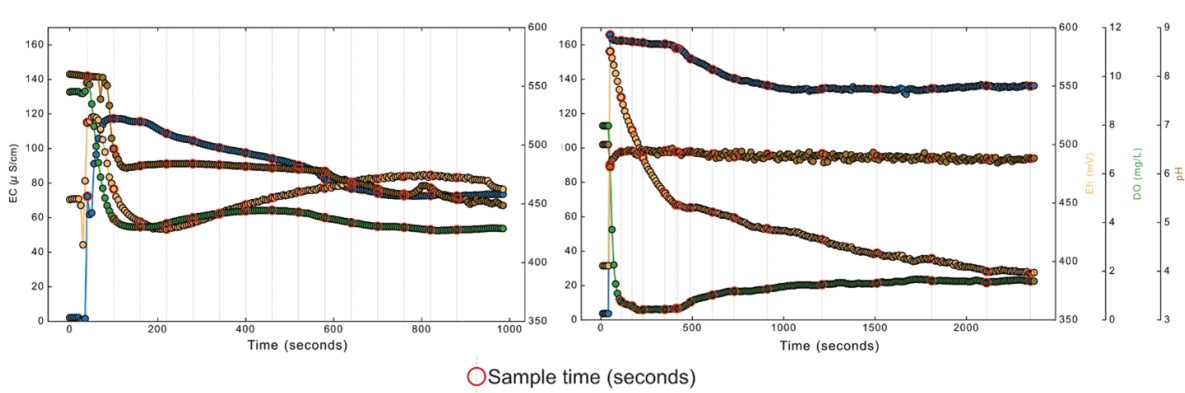

**Figure 4.** High-frequency time series of the physico-chemical parameters recorded at GW5 on March 24th, 2021 (left) and August 2nd, 2022 (right).

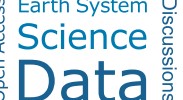

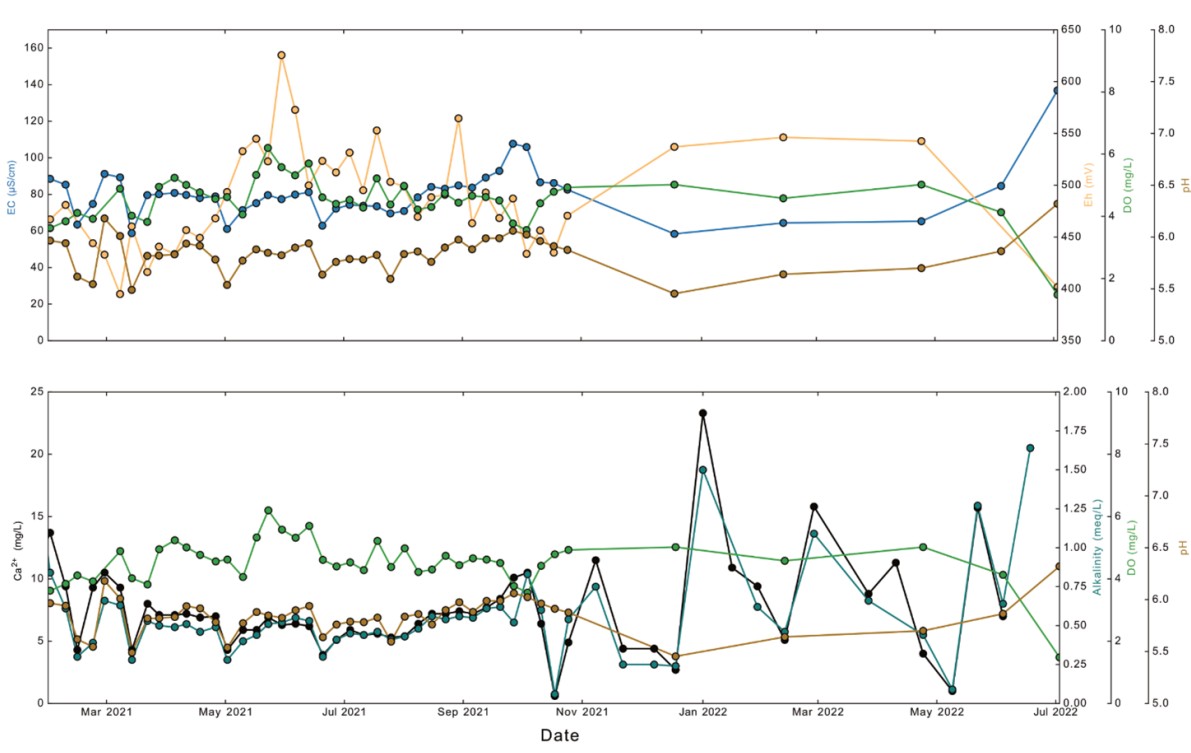

**Figure 5.** Time series of the physico-chemical parameters (top) and the related solute concentrations (bottom) at GW5, collected during the weekly sampling campaigns from March 2021 to June 2022.



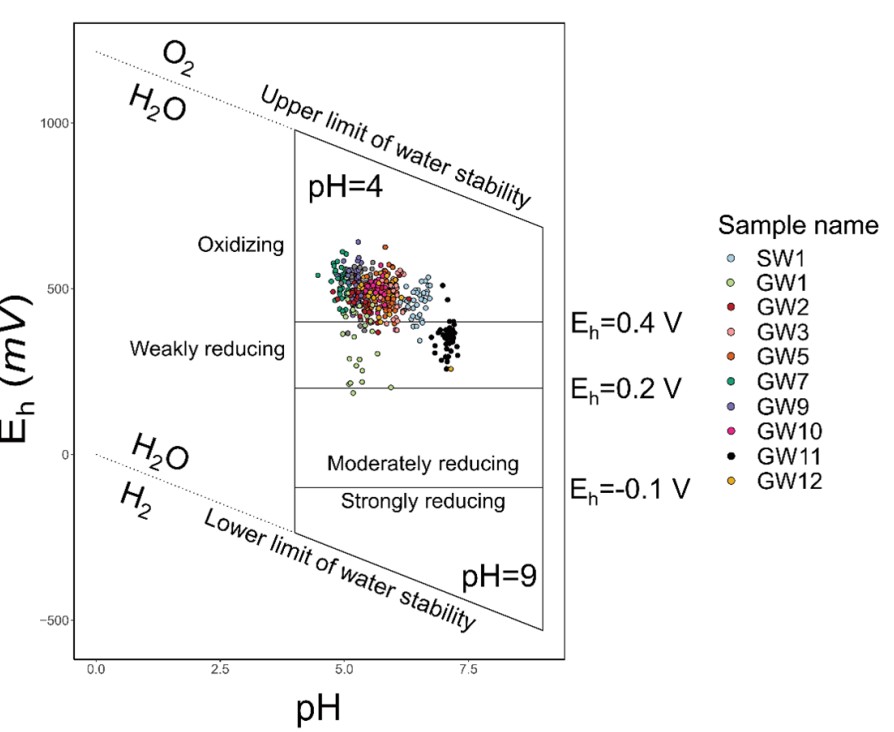

**Figure 6.** Eh-pH diagram depicting weekly in-situ field observations collected over the course of a year. The zones depicted on the graph are adapted from Zhang and Furman (2021). The upper and lower bounds represent the Eh - and pH stability ranges for water in natural environments. Adapted from Zhang and Furman (2021).

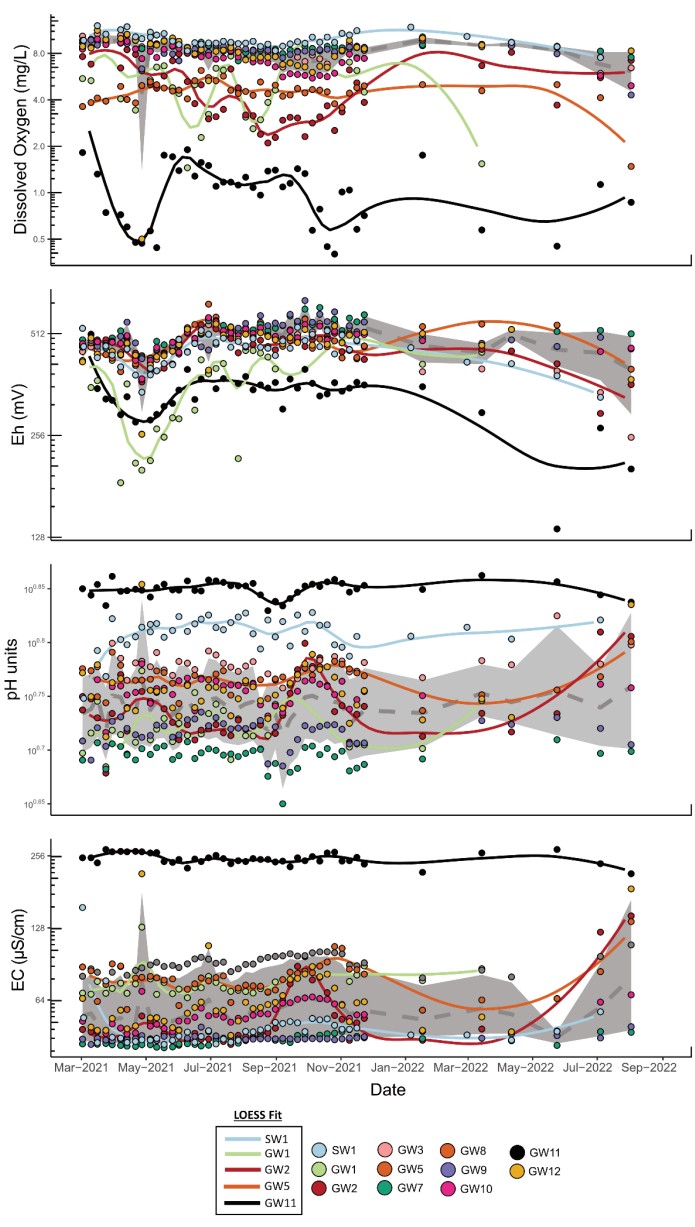

**Figure 7.** Time series of dissolved oxygen (DO), Eh, pH, and electrical conductivity (EC) from March 2021 to August 2022 for boreholes in the Weierbach Experimental Catchment. The grey dashed line and shaded area denotes the median and 5-95% bounds for the 'Shallow flowline' group, including GW3, GW7, GW9, GW10, and GW12. LOESS fits for SW1, GW1, GW2, GW5, and GW11 illustrate the variation in parameter trends across different locations within the catchment, highlighting differences between shallow and deep flowlines. GW11, located in the streambed, represents the deep flowline and the catchment's deepest point.



**Table 1.** Characteristics of the boreholes selected for the monitoring of the groundwater physico-chemical parameters during this study.

| ID | XWgs84 | Ywgs84 | Sampling Depth | Standpipe | Diameter | Screen |
|---|---|---|---|---|---|---|
| GW1 | 5.79676 | 49.83466 | 220 cm | 32 cm | 50 mm | 1 m depth to bottom |
| GW2 | 5.79623 | 49.83110 | 195 cm | 35 cm | 50 mm | 1 m depth to bottom |
| GW3 | 5.79721 | 49.82832 | 222 cm | 32 cm | 50 mm | 1 m depth to bottom |
| GW5 | 5.79715 | 49.83468 | 753 cm | 33 cm | 32 mm | 1 m depth to bottom |
| GW7 | 5.79465 | 49.83225 | 555 cm | 18 cm | 32 mm | 1 m depth to bottom |
| GW9 | 5.79713 | 49.83332 | 628 cm | 28 cm | 40 mm | 1 m depth to bottom |
| GW10 | 5.79706 | 49.83146 | 299 cm | 31 cm | 40 mm | 1 m depth to bottom |
| GW11 | 5.79709 | 49.83137 | 100 cm | 30 cm | 40 mm | no screen |
| GW12 | 5.79750 | 49.83220 | 305 cm | 40 cm | 40 mm | 1 m depth to bottom |





**Table 2.** Quantification limit (QL) of the chemical parameters considered in this study.

| | $SiO_2$ | $NH_4^+$ | $HPO_4^{3-}$ | $Na^+$ | $K^+$ | $Ca^{2+}$ | $Mg^{2+}$ | $Cl^-$ | $NO_3^-$ | $SO_4^{2-}$ | Al | Fe | Mn | Ba | Sr |
|---|---|---|---|---|---|---|---|---|---|---|---|---|---|---|---|
| | mg L$^{-1}$ | | | | | | | | | | $\mu$g L$^{-1}$ | | | | |
| QL | 0.1 | 0.01 | 0.15 | 0.1 | 0.02 | 0.2 | 0.1 | 0.1 | 0.04 | 0.1 | 4.0 | 1.0 | 0.1 | 50 | 30 |