# Peer review of "One year of high frequency monitoring of groundwater physico-chemical parameters in the Weierbach Experimental Catchment, Luxembourg."

_Earth System Science Data, 2024_

## Author Response (AR1)

**Response to Reviewers & Summary of Revisions**

We sincerely appreciate the constructive feedback from the reviewers. Below, we provide a detailed response to each comment, along with specific manuscript revisions. Given that revisions were made in Overleaf, we are unable to provide track changes, but a compiled comparison report showing the changes is available.

**Response to Reviewer 1:**

The manuscript presents a comprehensive database on groundwater levels and physico-chemical properties with high frequency measurements and sampling in the small Weierbach catchment in Luxembourg. This type of data is fundamental for sustainable water resources management, environmental and public health protection and future planning in the face of climate change. In times of dynamic changes in natural environment, we need as much of this type of data as possible from different areas.

The description of the data base, the data acquisition methods and the possibilities for their use have been clearly and accurately drawn up. It is evident that the authors have applied rigorous data quality control criteria. And I have no doubt that this database can be a valuable and useful source of information for scientists.

However, these are not the first studies conducted with such high frequency. There are experimental catchments in Europe where similar measurements have been carried out for a longer time. Examples include the TERENO experimental catchments in Germany https://www.tereno.net/, including the Wüstebach catchment, which is relatively close to Weierbach and with similar environmental conditions. Sampling of groundwater and determination of its chemical properties also takes place there on a weekly basis. It would be good to refer to measurements in other areas; is the research described by the authors part of the current of European research, or does it contribute something more?

Secondly, I have my doubts that data collected for one year can be sufficient to determine 'catchment response to hydrological change', ' catchment seasonal variability', 'dynamics between shallow and deeper flowlines' and whether it is sufficient for any modelling. Each year is different and at least several years of data are needed to demonstrate any patterns. Perhaps some statements could be reworded or softened.

Thirdly, the authors extensively describe the usage and applications of the data, but what about the weaknesses or limitations of data usage? A paragraph or subsection discussing these limitations and how to address them would be helpful.

Thank you for your detailed feedback and constructive suggestions, please see below our response:

1. Contextualization with Other Studies and European Relevance: We appreciate your point regarding similar datasets, such as those from the TERENO network, including the Wüstebach catchment because this would put this new dataset in a more general/European framework and strengthen its value. Although weekly sampling of groundwater chemistry in European experimental catchments is not new, our study contributes to this body of work by offering in situ redox potential measurements directly from the borehole, allowing for real-time data collection. This approach, along with our specific focus on differentiating shallow and deep groundwater flow paths in an entire headwater catchment, provides finer temporal and spatial resolution. This data structure aligns with ongoing European research but adds to it by offering redox and biogeochemical dynamics across vertical profiles, aiming to reveal interactions within both shallow and deeper aquifer zones. We will add an additional paragraph highlighting literature review to contextualize the database within existing European research, highlighting both similarities and distinctions.

**Reviewer's Concern:**

The manuscript should better contextualize the dataset within similar European research efforts, particularly referencing the TERENO Wüstebach catchment and other long-term monitoring networks.

**Response & Revisions:**

We acknowledge this important point and have expanded our discussion to better contextualize the dataset within ongoing European Critical Zone research. Specifically, we now reference similar observatories such as TERENO, SoilTrEC, and Calhoun CZO and emphasize how our dataset contributes through in situ redox potential measurements and depth-specific hydro-biogeochemical monitoring.

**(Line 50 to 61 in the revised manuscript)**

Our study complements existing Critical Zone research efforts across Europe and beyond. Various observatories, such as the TERENO Wüstebach Catchment in Germany, have advanced our understanding of soil moisture dynamics, land-atmosphere exchange, and hydrological processes through dense sensor networks (Heistermann et al., 2021). Similarly, the SoilTrEC network of observatories across Switzerland, Austria, Greece, and the Czech Republic focusses on soil transformation, biogeochemical cycling, and the impacts of land use (Guo and Lin, 2016). The Calhoun CZO in the USA provides insights into deep soil biogeochemical processes, particularly carbon cycling and hydrological connectivity (Lee et al., 2023). Our dataset from the Weierbach Experimental Catchment (WEC) provides high-resolution temporal data on groundwater physicochemical parameters, with a particular focus on redox dynamics and pH fluctuations. Although many observatories investigate both surface and subsurface processes, high-frequency monitoring of subsurface redox dynamics zonation remains limited. Our study offers initial insight into vertical biogeochemical dynamics within the subsurface, contributing to a better understanding of redox zonation, subsurface hydrological pathways, and the interplay between hydrological variability and biogeochemical processes in temperate forested catchments.

2. Duration Collection of Data and Interpretation Limitations: We acknowledge that the one-year dataset is limited in capturing long-term trends, including inter-annual variability in catchment responses, seasonal changes, and flowline dynamics. The first year of monitoring coincided with one of the wettest years in the last decade, which may influence the representativeness of the data. However, in the two years of monitoring, we observed hydrological states that reflect most of the conditions recorded in this catchment over the past 20 years. This suggests that the dataset provides valuable insight into typical catchment-scale dynamics of the measured physico-chemical parameters. We will adjust the manuscript to tone down any claims of comprehensive conclusions and instead highlight the dataset's potential as a foundation for understanding key processes. The continuation of biweekly sampling and the existing 10-year dataset will help contextualize these findings and support more robust long-term analyses.

**Reviewer's Concern:**

A one-year dataset may be insufficient to determine long-term catchment responses, seasonal variability, and hydrological flowline dynamics.

**Response & Revisions:**

We agree and have softened our conclusions, highlighting that our dataset represents a preliminary step towards understanding these processes. We now clarify that although the monitoring lasted one year, it captured contrasting hydrological extremes (wettest and driest periods) and is complemented by ongoing bi-weekly monitoring and a 10-year dataset.

**Revised text in Lines 186-192:**

We acknowledge that a one-year dataset is limited in capturing long-term trends, including interannual variability in catchment responses, seasonal changes, and flowline dynamics. The first year of monitoring coincided with one of the wettest years in the last decade, which may influence the representativeness of the data. However, over two years of monitoring, we observed hydrological states reflecting most conditions recorded in this catchment over the past 20 years. This suggests that the dataset provides valuable insight into typical catchment-scale dynamics of the measured physico-chemical parameters. We have adjusted the manuscript to tone down any claims of comprehensive conclusions and instead highlight the dataset's potential as a foundation for understanding key processes. The continuation of bi-weekly sampling and the existing 10-year dataset will help contextualize these findings and support more robust long-term analyses.

3. Limitations of the Dataset and Potential Weaknesses: We agree that discussing the limitations of our data, particularly its one-year span, would be valuable for future users. Aside from duration, **technical issues** such as potential probe drift due to temperature fluctuations may impact readings for redox and dissolved oxygen, and we acknowledge this as a possible limitation of on-site measurement techniques. To mitigate this, we have applied temperature correction protocols and intend to include notes on any observed drift in the supplementary dataset. Additionally, as you suggested, we will include a section discussing these limitations and outline how we aim to address them in the future.

**Reviewer's Concern:**

The manuscript extensively discusses the dataset's applications, but does not mention its limitations.

**Response & Revisions:**

We have added a new limitations section, discussing:

- 4. One-year monitoring timeframe (limiting inter-annual patterns)
- 5. Potential sensor drift (redox and DO affect by temperature changes)
- 6. Limited spatial coverage of certain boreholes
- 7. Plans for future improvements

**Newly added limitations section (Lines 280-290):**

Although the dataset provides high-resolution insight into groundwater biogeochemistry, some limitations should be considered. The monitoring period of one year, while covering a range of hydrological conditions, does not fully capture long-term inter-annual variability. Future research should incorporate extended monitoring to better understand long-term trends. Additionally, temperature fluctuations may cause probe drift in redox and dissolved oxygen measurements, although temperature correction protocols were applied. Some boreholes have limited depth coverage, which may influence the representation of deeper groundwater dynamics. Ongoing bi-weekly sampling and a complementary 10-year dataset will help mitigate these limitations and provide a broader context for long-term analyses.

Thank you again for your valuable insights, which will help strengthen our manuscript.

**Specific comments:**

**Reviewer's concern**

**Lines 43-45: This sentence requires references.**

"Despite widespread recognition of the shallow and deep hypothesis, direct data support of the hypothesis using measured stream chemistry in conjunction with subsurface water chemistry at different depths is rare."-(Steward et al., 2022)

Stewart, B., Shanley, J. B., Kirchner, J. W., Norris, D., Adler, T., Bristol, C., et al. (2022). Streams as mirrors: Reading subsurface water chemistry from stream chemistry. Water Resources Research, 58, e2021WR029931. https://doi.org/10.1029/2021WR029931

**Line 94: I suggest adding 'level' between groundwater and monitoring.**

**Will do -done**

Lines 121-127: It is not clear to me whether the field measurements were conducted directly in the borehole or if the water was first collected and then measured. If it's the latter, into what kind of container was the water collected, was the measurement taken immediately after collection or with some delay, and what was the volume of such a sample?

We used a **peristaltic pump** connected to a flow-through cell system with four cells (two in series, and two in parallel for pH and ORP to avoid interference). Groundwater was pumped until EC stabilization, which indicated borehole homogenization. The system was then **sealed at both ends** to prevent external influences, allowing a 20-minute equilibration period before final readings were taken directly from the closed system.

**Response & Revisions:**

We have rewritten the text to make it more clear, please see below the revised text (Line 140 to 147):

"Before measuring the parameters and sampling a well, the system was flushed with one litre of demineralised Milli-Q water to remove potential contaminants. The probes were then allowed a 5-minute period to record the EC reading, which ranged from 0.1 to 1.7  $\mu$ S/cm. Subsequently, the borehole was pumped and the four parameters were continuously recorded at increments of one minute. A groundwater sample was collected when the EC reading remained stable for several minutes, under the assumption that the borehole demonstrated homogeneity. Measurements were conducted in-line to minimise atmospheric contact and temperature changes. After collecting the water sample, a 20-minute equilibration period was implemented, in which the flow-through system was sealed and the probes were allowed to equilibrate with the solution in the cells. The final readings for EC, DO, ORP, and pH were then obtained. The physico-chemical database consists of three subsets:"

**Reviewer's concern**

Line 186: Does the provided database from one-year monitoring campaign truly allow us to address the mentioned problems? Isn't there a need for longer measurement series? Perhaps the authors should soften this statement somehow, for example, by discussing the potential of the data, etc.

We agree that a one-year dataset may not capture the full range of temporal variability. This highfrequency, one-year monitoring campaign was designed to add resolution to key parameters that were suspected to influence biogeochemical processes in the catchment. While the timeframe was constrained by the project's budget and duration, we continued bi-weekly measurements after this campaign, and a separate 10-year bi-weekly dataset is also available. This longer-term dataset will contribute to future analyses and provide broader context.

The primary aim of the one-year dataset is to establish redox ranges and chemistry profiles for both shallow and deeper groundwater in this experimental catchment, offering a foundation for understanding the catchment's biogeochemical dynamics.

Thank you for the suggestion; we will revise the text to reflect these points.

**Response & Revisions:**

This has now been revised in the text to reflect what Reviewer 1 has noted (Line 196 to 210):

"Our dataset provides an opportunity to explore the contributions of shallow and deep flow paths during borehole pumping. We propose that purging a borehole until the extracted water stabilizes reflects the natural mixing processes that occur when groundwater discharges into streams. By conducting pumping tests under varying hydrological conditions and at multiple locations, we aim to investigate whether this approach can provide insights into the depth and spatial distribution of redox zones, which we hypothesize to be influenced by vertical permeability. Since the succession of electron acceptors drives the stratification of the redox zone (McMahon and Chapelle, 2008), the redox potential of extracted water (Eh) may offer clues about the relative contributions of different flow paths. Although our dataset does not provide a direct map of redox zones, it offers high-resolution observations that may help assess their role in shaping groundwater chemistry and, by extension, stream composition. The following section outlines how this dataset can potentially address key gaps in understanding redox processes, their influence on c-Q relationships, and the broader implications for hydrobiogeochemical research."

**Reviewer's concern**

**Lines 187-188: The authors mention previous studies, but they do not specify which studies they are referring to. It would be helpful to include references to these studies here.**

This section has been revised to take a more cautious approach regarding the novelty of the dataset. We now acknowledge that many similar datasets exist across various Critical Zone observatories. However, rather than listing specific references—given their abundance and easy accessibility through databases like Scopus—we clarify that our study focuses on a specific aspect of groundwater behaviour that is less well-documented in comparable studies.

Lines 200-202: Same as previous. Provide references to these previous studies.

This has been fixed, see Table 1

**Lines 220-221: Provide some exaples/references**

This has been fixed, see Table 1

Lines 231-233: Same as before. The references are missing.

We have further revised the text to take a more cautious approach regarding the novelty of the dataset, instead emphasizing how it contributes to the existing body of research. The specific revisions are detailed in Table 1.

**Response & Revisions:**

**Table 1: Summary of specific comments and the subsequent revisions**

| Reviewer's Comment     | Response & Action Taken                                                               |
|------------------------|---------------------------------------------------------------------------------------|
| Lines 43-45: Add       | Added Stewart et al. (2022).                                                          |
| references for stream  |                                                                                       |
| chemistry interactions |                                                                                       |
| Line 94: Add "level"   | Fixed.                                                                                |
| between groundwater    |                                                                                       |
| and monitoring         |                                                                                       |
| Lines 121-127 Clarify  | Clarified peristaltic pump setup and flow-through system.                             |
| in situ measurement    |                                                                                       |
| procedure              |                                                                                       |
| Lines 187-188 200-     | Added missing citations or rewrote section that were not accurate                     |
| 202 220-221 231-233    |                                                                                       |
| Missing references     | Lines 187-188                                                                         |
| Missing references     | Ellies 107-100
Revised line 187-102 (new line 211 to 216):                  |
|                        | Although provides studies have explored a $\Omega$ relationships and the              |
|                        | Although previous studies have explored c-Q relationships and the                     |
|                        | Innuence of various water sources of stream chemistry, this dataset                   |
|                        | provides high-resolution monitoring with a focus on capturing redox                   |
|                        | 2011e dynamics over space and time. By measuring four key parameters                  |
|                        | at weekly intervals, it offers a detanet perspective on these processes,              |
|                        | complementing existing datasets that often emphasise other                            |
|                        | biogeochemical aspects. While not entirely novel in its approach, the                 |
|                        | dataset adds valuable insight into the spatio-temporal variability of redox           |
|                        | conditions in forested catchments.                                                    |
|                        |                                                                                       |
|                        | Lines 200-202                                                                         |
|                        | Quantifying Redox Reactions - Arora et al. (2022), (Lazareva et al.,                  |
|                        | 2022).                                                                                |
|                        |                                                                                       |
|                        | Lines 220-221                                                                         |
|                        | Lazareva et al. 245 (2022),                                                           |
|                        | Lines 004 000                                                                         |
|                        | Lines 231-233
The lines the mediane second to be referenced                        |
|                        | The lines the reviewer asked to be referenced:                                        |
|                        |                                                                                       |
|                        | "One of the dataset's unique strengths is its capacity to elucidate the               |
|                        | seasonal variability and flowline interactions within the catchment.                  |
|                        | Previous studies may lack the temporal granularity needed to capture                  |
|                        | these nuances adequately."                                                            |
|                        |                                                                                       |
|                        | It was addressed by revising the text: ( line 254 to 259 in current version ): |
|                        |                                                                                       |
|                        | "The dataset contributes to ongoing efforts to understand seasonal                    |
|                        | variability and flowline interactions within catchments, adding to the                |
|                        | work of previous studies that have monitored these dynamics. While                    |
|                        | high-spatiotemporal-resolution datasets exist, this dataset provides                  |
|                        | complementary observations that may help identify shifts in flowline                  |
|                        | dominance under varying hydrological conditions. Its focus on capturing               |
|                        | redox zone development across different spatial and temporal scales                   |

|                                                                      | offers an additional perspective on groundwater flow and chemistry in forested catchments." |
|----------------------------------------------------------------------|---------------------------------------------------------------------------------------------|
| Figure 5: SO 4 and NO 3 concentrations missing | Fixed incorrect figure legend labels.                                                       |
| Table 1: Standardizecoordinate notation                              | Standardized to WGS format.                                                                 |

**Reviewer's concern**

Lines 248-249: What do you mean by 'extends beyond conventional monitoring efforts'? In other experimental catchments, groundwater measurements have also been conducted on a weekly basis for several years. It would be good to reference these studies and clearly state what is new about your measurements.

By "extends beyond conventional monitoring efforts," we intended to emphasize that, at the **catchment scale**, our high-frequency, on-site monitoring of EC, DO, ORP, and pH captures both spatial and **temporal dynamics** in groundwater biogeochemistry. This dataset focuses specifically on redox dynamics across both shallow and deep groundwater zones, providing a depth-specific profile.

We will conduct a brief literature review to confirm whether this dataset is novel in this regard and will update this section with references to similar studies as appropriate.

**Response & Revisions:**

**Line 287 to 294**

"The one-year monitoring campaign conducted within the WEC provides a dataset that contributes to ongoing efforts to understand the interactions between hydrological and biogeochemical processes in shallow groundwater. While similar datasets exist, this study adds to the existing body of work by offering high-resolution observations that may help to explore seasonal variability, flowline interactions, and redox zone development. The dataset does not provide a comprehensive understanding of these processes, but offers empirical data that could support future research into the influence of redox reactions on pH dynamics and ion exchange in forested catchments. These insights, though limited to a one-year period, provide a basis for considering key processes that can influence the hydrobiogeochemical behaviour of the catchment, as illustrated by the following aspects."

**Reviewer's concern**

Line 260: I would rather write about the potential significance of this database, provided that the measurements continue, because can a one-year measurement campaign really address these issues?

We agree that a one-year measurement campaign may be limited in capturing long-term variability. This dataset is intended as an initial step to understand seasonal dynamics and biogeochemical interactions, providing baseline data for key parameters. We agree that its potential significance will increase with ongoing monitoring, allowing us to build on these insights to better address long-term catchment processes.

In the revised text, we will adjust our language to emphasize the preliminary nature of these findings and the added value of continuing measurements over time.

**Response & Revisions:**

**Line 293:**

"These insights, though limited to a one-year period, provide a basis for considering processes that can influence the hydrobiogeochemical behaviour of the catchment, as illustrated by the following aspects."

**Reviewer's concern**

Figure 5. The text refers to Figure 5, which is supposed to show graphs of SO4 and NO3 concentrations, but they are missing from the figure. Additionally, there is no need to repeat the DO and pH graphs on the lower plot, as they are already present on the upper one.

Thank you for noticing this. The labels in the legend of the lower plot in Figure 5 are incorrect; they should refer to  $SO_4$  and  $NO_3$  rather than DO and pH. We will correct this to accurately reflect the data shown in the figure.

Response & Revisions: This has been fixed

Technical corrections:

Reviewer's concern Line 24: remove dot after "reactive rock formations"

Will do.

Response & Revisions: **Done**

**Reviewer's concern**

Figure 4: I would eliminate the black outlines of the circles, as it is difficult to distinguish the colours with such a high density of points.

Will try this and potentially other solutions to address the problem you have pointed out.

**Response & Revisions:**

I have removed the black outlines as suggested but found that it did not enhance the clarity of the figure. After careful consideration, I have decided to retain the original version. However, if the editor believes that this adjustment is still necessary, I am happy to provide an alternative version of the figure upon request.

**Reviewer's concern**

Table 1: Please standardize the notation of the coordinates in either uppercase or lowercase letters (WGS or wgs).

Will do.

Response & Revisions: Done

**Response to Reviewer 2:**

**Reviewer's concern**

I would only recommend changing certain colours in figures, particularly yellow.

**Response & Revisions:**

There is no yellow in the graphs; the colour used was orange. Any perceived discrepancy may be due to the smaller graph size in the preprint, which could have affected its appearance.

**Response to Reviewer 3:**

**Reviewer's Concern:**

In the section 2, a detailed description of the forest vegetation (tree-species distribution, percentage of the forest cover within the catchment, tree density, etc.) is incomplete. This detailed description would allow the reader to get a clearer image of the study area in which the groundwater was monitored. A forest map would be also helpful.

Lines 60 -65: You should provide here the scientific name for all the tree and shrub species mentioned in the text.

**Response & Revisions:**

We have revised the text in accordance with the reviewer's comments. Please find the revised text below. We did not find it necessary to include a forest map and have instead referred the reader to relevant literature that provides a detailed description of the information in question.

**Line 79 to 84**

"The forest structure of the Weierbach Experimental Catchment (WEC) is shaped by past and current management practices.

Oak trees are evenly distributed, while beech tree density increases from the plateau to the footslope. The primary plateau80

hosts a dense mixed forest composed of 78% Fagus sylvatica and 22% Quercus petraea × Q. robur (Fabiani et al., 2022). The understory is dominated by Vaccinium myrtillus, while the riparian zone features various short plants (Martínez-Carreras et al., 2015). The secondary plateau is primarily composed of Picea abies and Pseudotsuga menziesii. A detailed analysis of forest

structure, including tree density and ecohydrological interactions, can be found in Fabiani et al. (2022, 2024)."

**Reviewer's Concern:**

Figure 1 (a): Contour lines are drawn on the map, but without altitudinal values; at least in the case of the main contour lines (5 m), the altitudinal values should be provided.

**Response & Revisions:**

I have revised the figure caption to specify the elevational range of the catchment (450–500 m):

"Figure 1. Presentation of the Weierbach Experimental Catchment: (a) Topography with water sampling locations, including contour lines representing 5-meter elevation intervals (altitudinal values not labeled; catchment elevation ranges from 450 to 500 m), and (b) regolith structure. The black contour line in map (a) marks the chemical weathering front basement, derived from ERT surveys and core drillings (Gourdol et al., 2021)."

**Conclusion**

We hope this report facilitates the finalization of the manuscript. If there are any further questions or uncertainties, please do not hesitate to reach out. We would be happy to provide any additional clarifications or revisions as needed.